evolution/molecular biology/genetics

*Dendrogale murina*, northern smooth-tailed treeshrew, *Tupaia everetti*, *Urogale*, Mindanao treeshrew, colour vision

**Author for correspondence:**
Amanda D. Melin
e-mail: amanda.melin@ucalgary.ca

# Opsin genes of select treeshrews resolve ancestral character states within Scandentia

Gwen Duytschaever[1], Mareike C. Janiak[1,2], Perry S. Ong[3], Konstans Wells[4], Nathaniel J. Dominy[5,6] and Amanda D. Melin[1,2,7]

[1]Department of Anthropology and Archaeology, University of Calgary, Calgary, Alberta, Canada
[2]Alberta Children's Hospital Research Institute, Calgary, Alberta, Canada
[3]Institute of Biology, University of the Philippines Diliman, Quezon City, Philippines
[4]Department of Biosciences, Swansea University, Wales, UK
[5]Department of Anthropology, Dartmouth College, Hanover, NH, USA
[6]Department of Biological Sciences, Dartmouth College, Hanover, NH, USA
[7]Department of Medical Genetics, University of Calgary, Calgary, Alberta, Canada

 GD, 0000-0002-6559-3922; MCJ, 0000-0002-7759-2556;
KW, 0000-0003-0377-2463; NJD, 0000-0001-5916-418X;
ADM, 0000-0002-0612-2514

Treeshrews are small, squirrel-like mammals in the order Scandentia, which is nested together with Primates and Dermoptera in the superordinal group Euarchonta. They are often described as living fossils, and researchers have long turned to treeshrews as a model or ecological analogue for ancestral primates. A comparative study of colour vision-encoding genes within Scandentia found a derived amino acid substitution in the long-wavelength sensitive opsin gene (*OPN1LW*) of the Bornean smooth-tailed treeshrew (*Dendrogale melanura*). The opsin, by inference, is red-shifted by *ca* 6 nm with an inferred peak sensitivity of 561 nm. It is tempting to view this trait as a novel visual adaptation; however, the genetic and functional diversity of visual pigments in treeshrews is unresolved outside of Borneo. Here, we report gene sequences from the northern smooth-tailed treeshrew (*Dendrogale murina*) and the Mindanao treeshrew (*Tupaia everetti*, the senior synonym of *Urogale everetti*). We found that the opsin genes are under purifying selection and that *D. murina* shares the same substitution as its congener, a result that distinguishes *Dendrogale* from other treeshrews, including *T. everetti*. We discuss the implications of opsin functional variation in light of limited knowledge about the visual ecology of smooth-tailed treeshrews.

# 1. Introduction

Treeshrews are small, squirrel-like mammals with a broad distribution from India and southern China through most of Southeast Asia (figure 1a). They comprise a single order, Scandentia, in which two families are recognized: Ptilocercidae, containing a nocturnal species, *Ptilocercus lowii,* and Tupaiidae, containing 22 diurnal species [4] in four traditional genera (*Dendrogale, Anathana, Urogale* and *Tupaia;* figure 1b). Scandentia is nested in the superordinal group Euarchonta together with Dermoptera and Primates (figure 1b), an affinity that invites the use of treeshrews as a model system for studying a wide range of human disorders, including myopia [6]. Treeshrews are also viewed as 'living fossils' [7] and therefore practical models or ecological analogues of ancestral and stem primates [8–14]. Accordingly, Melin *et al*. [15] sequenced the opsin genes of Bornean treeshrews to contextualize the origins of high-acuity colour vision in primates. In *Dendrogale melanura*—the earliest branching tupaiid in their sample—they found an amino acid substitution ($_A180_S$) that translates into a relatively red-shifted long-wavelength sensitive (LWS) opsin. Given that opsins are sensitive to ecological selective pressures [16], it is tempting to interpret this genotype as a novel visual adaptation; however, the full extent of its derivation in Tupaiidae is uncertain. To resolve this uncertainty, data are needed from treeshrew species outside Borneo.

Here we fill two voids by focusing on the only extant congener of *D. melanura*, the northern smooth-tailed treeshrew (*Dendrogale murina*) and the Mindanao treeshrew, a species described as *Tupaia everetti* in 1892 and elevated to a monotypic genus (*Urogale*) in 1905 on the basis of distinguishing morphological traits [17]. This latter nomen prevailed for a century until mounting molecular evidence favoured the subsumption of *Urogale* into *Tupaia* [1,2,18]. *Tupaia everetti* is therefore the senior synonym of *U. everetti*, and re-recognition of *T. everetti* is spreading in the literature [4,5]. Setting this taxonomic reversal aside, *T. everetti* holds interest because it is the earliest branch in crown *Tupaia*, splitting prior to the diversification of *Tupaia* across Southeast Asia (figure 1b). It is therefore crucial for pinpointing the origin of differential opsin sensitivities in Tupaiidae.

Accordingly, we examined the sequences of exons known to determine the spectral tuning of the LWS opsin gene (*OPN1LW*) as well as the short-wavelength sensitive opsin gene (*SWS1, OPN1SW*), which has been subject to different selective pressures among taxa within Scandentia and more generally across Euarchonta [15]. We then compared the sequences of *D. murina* and *T. everetti* to those of other treeshrews and primates in order to reconstruct the ancestral character states of opsin genes at two nodes within Tupaiidae.

# 2. Material and methods

## 2.1. Study species and sample collection

*Dendrogale murina* is one of two recognized species in the genus (figure 2a). It has a wide distribution throughout Vietnam, Thailand and Cambodia [19] in contrast to its congener (*D. melanura*), which is endemic to montane Borneo [20]. It is also more flexible ecologically, with records ranging from lowland plains to 1500 m across a wide range of habitat conditions, from evergreen forest (at varying stages of degradation), to mixed deciduous forest, to secondary bamboo fields lacking any dicotyledonous canopy, to streamside tangles in rocky savannah [19]. It not only uses the under- and mid-storeys, but also enters the canopy; recent observations come primarily from understorey tangles, especially of bamboo, almost exclusively 30–300 cm above ground level [19]. In Thailand, it has been observed on the branches of fruiting trees [19].

*Tupaia everetti* is the sole scandentian to inhabit the Mindanao Faunal Region, Philippines (figure 2b; [17,21]). It has a widespread distribution on Mindanao, preferring montane and mossy forests between 750 and 2250 m a.s.l., but it is also found at lower elevations on neighbouring islands [5]. Limited observations and stomach content analysis indicate a diurnal terrestrial niche and a mixed diet of ground-dwelling insects, other arthropods, fruits and other plant material [22,23]. It is readily differentiated from other tupaiids by its even-haired round tail, elongated snout and large, canine-like second incisors [17]; it is also the largest treeshrew, with captive adults ranging from 270 to 410 g (mean: 314 g; $n = 7$ [24]) and 190 to 342 g (mean: $276 \pm 8.0$ g; $n = 10$ males; mean: $252 \pm 6.3$ g; $n = 18$ females [25]). Wild-caught specimens are equally large (range: 235–315 g; $n = 2$ males [26]), but Heaney reported smaller values (mean: 190 g; $n = 3$; cf. Sargis [27]). Data are limited, but specimens

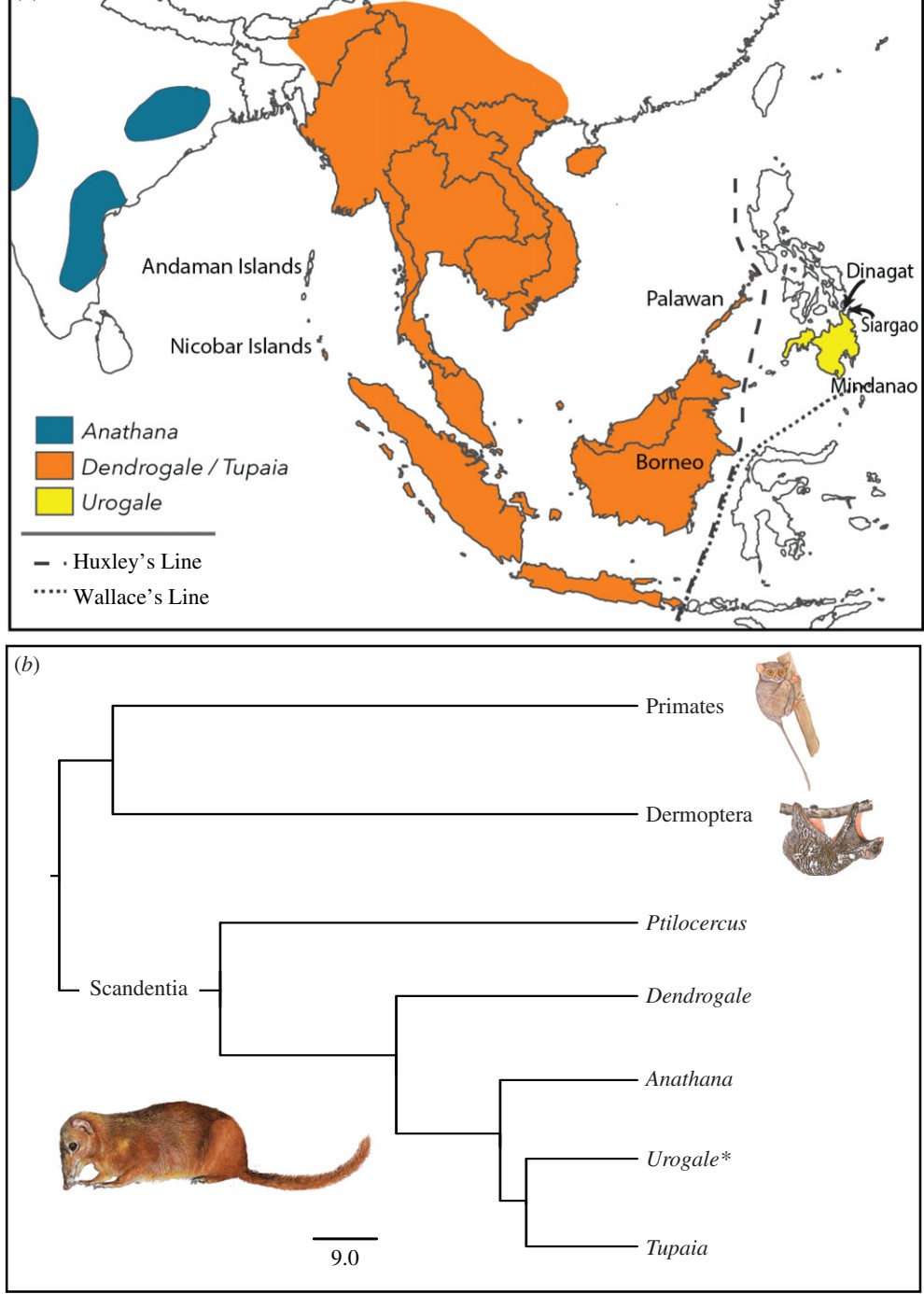

**Figure 1.** (a) Approximate distribution of treeshrew genera in the family Tupaiidae. *Dendrogale* is broadly sympatric with *Tupaia* in northern Borneo and Indochina. Huxley's Line generally corresponds to the edge of the Asian continental shelf and separates oceanic islands from landbridge islands in the Philippines. Redrawn from Roberts *et al*. [1]. (b) Phyletic relationships of genera historically recognized in the order Scandentia, which is sister to Dermoptera and Primates (Primatomorpha) in the superordinal group Euarchonta [2,3]. *Recently, *Urogale* has been subsumed into the genus *Tupaia* [2,4,5]. Original artwork by Priscilla Barrett, reproduced with permission.

of *T. everetti* from Dinagat and Siargao are smaller than those from Mindanao [28], raising the possibility of cryptic speciation on these islands [29].

We extracted DNA from the muscle tissues of museum specimens. The specimen of *D. murina* is accessioned in the University of Alaska Museum (catalogue no. UAM 103000). It was a wild-caught male from the Seima Biodiversity Conservation Area, Mondulkiri, Cambodia. Two specimens of *T. everetti* are housed at the Vertebrate Museum, Institute of Biology, University of the Philippines

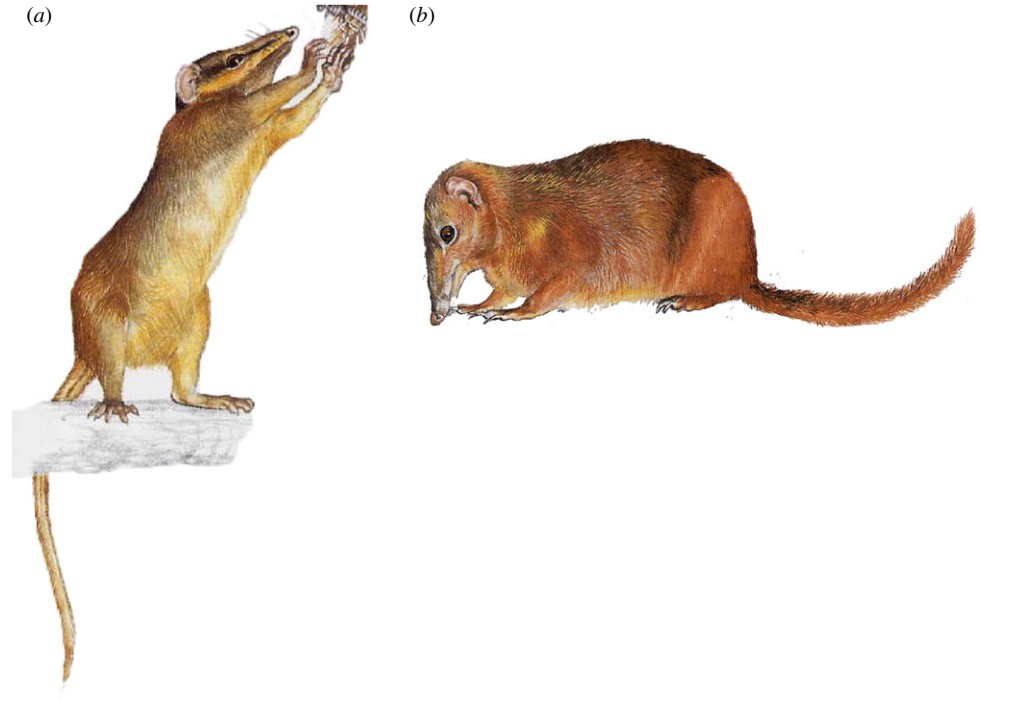

**Figure 2.** Illustrations of study species: (*a*) the Northern smooth-tailed treeshrew, *Dendrogale murina* and (*b*) the Mindanao treeshrew, *Tupaia everetti*. Original artwork by Priscilla Barrett, reproduced with permission.

Diliman, Quezon City (catalogue nos. PNM7496 [adult male] and PNM7497 [juvenile male]). The animals were wild-caught in the Mt Apo Natural Park, Barangay Agco, Kidapawan City, Cotabato, Mindanao. Muscle tissues were biopsied in the field and stored in 99% ethanol.

## 2.2. DNA extraction, amplification and sequencing

Genomic DNA was extracted from muscle tissues using a DNeasy Blood and Tissue Kit (Qiagen) following the manufacturer's instructions. Amino acids at 10 sites on exon 1 of the *OPN1SW* gene determine the spectral tuning of the opsin, with three sites—*86, 90* and *93*—primarily governing sensitivity in the violet-blue (400–450 nm) region of the light spectrum [30–32]. The $\lambda_{max}$ of the opsin encoded by *OPN1LW* is determined by five amino acid sites spanning exons 3–5 [33]. Three of these sites are variable in primates (exon 3: 180; exon 5: 277, 285; [34,35]); among treeshrews, genotypes at sites 180, 277 and 285 have recently been reported, with variation between taxa noted at site 180 [15]. We obtained partial opsin sequences for *D. murina* and *T. everetti* by amplifying exons 1, 2–3, 4 and 5 of the *OPN1SW* and exons 3 and 5 of the *OPN1LW*. Although the tuning sites of *OPN1SW* are located on exon 1, variation in gene functionality occurs within Scandentia, and we sequenced additional exons to rule out indels or premature stop codons leading to gene pseudogenization [15].

Polymerase chain reactions (PCRs) were conducted in 25 µl volumes containing 1× KAPA HiFi Readymix (Kapa Biosystems Inc., USA), 1.0 mM each of the forward and reverse primers (electronic supplementary material, table S1) and 200 ng template DNA. Molecular grade water was used as a negative control in all reactions. Thermocycler parameters were as follows: 3 min initial denaturation at 98°C; 35 cycles of 10 s denaturation at 98°C, 30 s annealing at either 58°C (for *OPN1SW* exons 1 and 4) or 60°C (for *OPN1SW* exons 2–3 and 5 and *OPN1LW* exons 3 and 5) and 30 s extension at 72°C, followed by a 5 min final extension at 72°C.

Amplification of target sequences was confirmed on a 1.5% agarose gel in 1 X Tris Borate EDTA (TBE) buffer. PCR products were purified using EXOsap-IT (Affymetrix, USA) following the manufacturer's protocol. If non-target sequences were also amplified, target amplicons were excised and purified using the Purelink Gel Extraction kit (Life Technologies Inc.). Purified PCR products were directly Sanger sequenced on sense and antisense strands at the University of Calgary Core DNA Sequencing Facility (Faculty of Medicine, University of Calgary, Alberta, Canada) using an Applied Biosystems 3730×l 96 capillary DNA Analyzer.

## 2.3. Data analysis

We assembled and edited *OPN1SW* and *OPN1LW* sequences in Geneious v. 10.0.3 (Biomatters) using the Clustal W function with manual refinement. We aligned them to opsin sequences from *Homo sapiens* and published treeshrew species: *Ptilocercus lowii, Dendrogale melanura,* and six species of *Tupaia* (*T. minor, T. belangeri, T. tana, T. montana, T. longipes* and *T. gracilis*; electronic supplementary material, table S2). We translated the coding regions into amino acid sequences, which we used to infer colour vision phenotype. Site numbers in our alignments correspond to the position of the amino acid in the human SWS1 and M/LWS pigments. We used the Protein Variation Effect Analyzer (http://provean. jcvi.org/; PROVEAN) tool to predict the functional implications of other nonsynonymous mutations [36,37]. PROVEAN predicts deleterious polymorphisms, but the functional effect of nonsynonymous mutations that change the amino acid will be predicted neutral when the properties of the amino acid do not change drastically (e.g. due to hydrophobicity). We tested for evidence of purifying or positive selection in *OPN1SW* and *OPN1LW* using codeml free-ratio branch models implemented in phylogenetic analysis by maximum likelihood (PAML) [38]. Outgroups for PAML analyses were two primates with functional opsin genes, *Alouatta palliata* (*OPN1SW*: AH005790.1; *OPN1LW*: AB809459.1) and *Tarsius bancanus* (*OPN1SW*: AB111463.1 and *OPN1MW*: AB675927.1) (electronic supplementary material, table S2). We also used codeml site models to test whether any sites, including the tuning sites are under positive selection. We did not include *P. lowii* in the PAML analysis of the *OPN1SW* gene due to low sequence coverage. This nocturnal species has a predicted *OPN1SW* pseudogene of ancient origin and relaxed selection pressures have previously been reported [15].

# 3. Results

We successfully sequenced partial exons 1–5 of *OPN1SW*, and exons 3 and 5 of *OPN1LW* for the *D. murina* and *T. everetti* specimens (electronic supplementary material, figure S1). Among scandentian sequences, the mean amino acid divergence (mean amino acid difference per sequence/total amino acids analysed, s.e. over 1000 bootstrap replicates) for *OPN1SW* was 1.02% divergence (3.524/347, s.e. = 0.822). The amino acid divergence for *OPN1LW* partial sequences was 0.96% divergence (1.311/ 136, s.e. = 0.567).

## 3.1. *OPN1SW* evolution

Relative to *T. everetti* and other *Tupaia* species, *D. murina* has four derived nonsynonymous variants causing amino acid differences: $_G111_A$ and $_V119_C$ in exon 1, $_T159_A$ in exon 2, and $_S224_T$ in exon 3 (electronic supplementary material, figure S1). The first two mutations are shared with *D. melanura*, but missing sequence data for *D. melanura* prevent assessing whether the last two are shared or unique to *D. murina*. At site 322, the *OPN1SW* amino acid sequence of *D. murina* has retained the ancestral state Cys322, relative to its congener, which has a derived nonsynonymous variant Phe322. All these nonsynonymous mutations are predicted to be neutral using PROVEAN, scores = −1.484 (site 111), 2.346 (site 119), −0.341 (site 159), 1.773 (site 224) and −1.903 (site 322); cut-off = −2.5). *Tupaia everetti* differs from *Tupaia* species in possessing L rather than Q at site 28. This may be the ancestral condition for Tupaiidae and possibly for Scandentia, as it is shared with *D. melanura*. Data at these sites from *P. lowii* would help to further resolve this but are presently unavailable. There are two derived substitutions unique to *T. everetti*: exon 1, $_A120_T$, exon 5, $_R323_K$ (electronic supplementary material, figure S1). All mutations were predicted to be neutral (PROVEAN scores = 4.742 (site 28), 0.977 (site 120) and −0.299 (site 323); cut-off = −2.5). Interestingly, a derived mutation in *T. longipes*, $_F115_V$, may be deleterious (PROVEAN score, −5.951). Overall, based on broad conservation of amino acids, including the main spectral tuning sites Tyr86, Ser90 and Val93 (exon 1), the $\lambda_{max}$ of the SWS1 opsin protein of both *D. murina* and *T. everetti* is predicted to be around 444 nm [15,30,32] (figure 3).

## 3.2. *OPN1LW* evolution

Two derived nonsynonymous mutations ($_A180_S$ and $_V182_I$, exon 3) are present in the *OPN1LW* gene sequence of *D. murina* and *D. melanura* compared to other treeshrews, indicating a shared origin in the last common ancestor of extant *Dendrogale* species (electronic supplementary material, figure S1). Nonsynonymous substitutions at site 180 and 182 were predicted to be neutral (PROVEAN respective

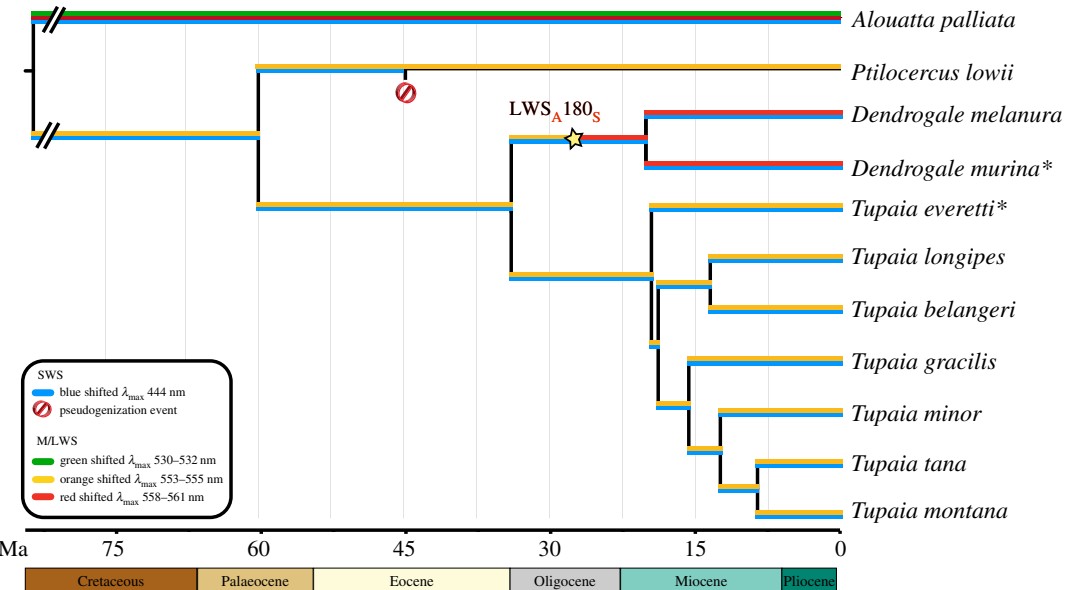

**Figure 3.** Phyletic relationships and divergence dates for primates and treeshrews were based on TimeTree [39] (accessed February 2019) and published estimates [2]. Branch colours correspond with the function and spectral tuning of opsin photopigments. Pseudogenization events are marked with a diagonally bisected circle. The inferred long-wave shift from orange to red sensitivity in the LWS opsin of the genus *Dendrogale* is marked with a star, along with the amino acid substitution responsible. The timing of this shift in sensitivity is unknown. Species sequenced in this study are indicated with an asterisk.

scores $= -0.302$ and $0.133$; cut-off $= -2.5$). However, site 180 is a known opsin tuning site (figure 3). We did not detect any derived nonsynonymous mutations in the *OPN1LW* sequence of *T. everetti*. Based on the three-site composition of SYT at sites 180, 277 and 285, *D. melanura* and *D. murina* are predicted to be long-wavelength shifted, and have a $\lambda_{max}$ value of *ca* 561 nm, as opposed to shorter $\lambda_{max}$ values for other treeshrews. We predict that the *T. everetti* LWS opsin protein (three-site composition of AYT) has a similar $\lambda_{max}$ to *Tupaia* and *Ptilocercus*, *ca* 555 nm.

## 3.3. Evaluation of selective pressures

The codeml branch models for *OPN1SW* indicated purifying selection acting on all species ($\omega < 0.312$), but supported a model of branch-specific differences in the strength of selection ($p = 0.045$, *Likelihood Ratio* ($LR$) $= 29.25$), showing stronger purifying selection acting on *D. murina* ($\omega = 0.0001$) than on *T. everetti* ($\omega = 0.312$). The codeml branch model did not find species-specific differences in the strength of selection acting on *OPN1LW*, indicating that the gene is under purifying selection ($\omega = 0.165$) in all examined species, including *D. murina* and *T. everetti*. We did not find evidence of positively selected sites in our codeml site models for either *OPN1SW* ($p = 1$, $LR = 0$) or *OPN1LW* ($p = 0.228$, $LR = 2.957$).

## 4. Discussion

We found that the short-wavelength sensitive opsin gene (*OPN1SW*) is intact in both *D. murina* and *T. everetti*. The amino acid composition of 10 spectral tuning sites, including the three most influential (Tyr86, Ser90, Val93) predicts a $\lambda_{max}$ of 444 nm, an inferred phenotype that unites *D. murina* and *T. everetti* together with every tupaiid examined to date [15]. In addition, we found that the long-wavelength sensitive opsin gene (*OPN1LW*) is functionally variable in the two study species. Despite PROVEAN's neutral prediction, *in vitro* site-directed mutagenesis of the LWS pigment has demonstrated that the single site mutation $_A180_S$ significantly shifts the $\lambda_{max}$ of the pigment [33]. Notably, *D. murina* shares the $_A180_S$ substitution with its congener *D. melanura* [15]. The LWS opsins of *Dendrogale* are therefore predicted to have a red-shifted $\lambda_{max}$ of 561 nm, whereas those of all other scandentians, including *T. everetti* (present results), are predicted to have a $\lambda_{max}$ of 555 nm.

Our results suggest that the colour vision of *T. everetti* is practically identical to that of other *Tupaia* species despite its relatively large body size and restricted distribution. This result is consistent with recent shared

ancestry [2], and it suggests comparable visual ecologies. Indeed, observations of habitat and resource use by these species speak to similar preferences for insects and fruit in the forest understorey [19,22,40,41]. It is possible that the 444/555 nm opsin combination is a 'multi-purpose' dichromacy with functionality across a wide range of understorey light conditions. Given the strict conservation of opsin genes in *Tupaia*, including the spectral tuning sites, it is rather surprising that *Dendrogale* is predicted to express a red-shifted LWS opsin, which suggests unique selective pressures on the colour vision of this lineage. The implications of this 6 nm red-shift are discussed below.

The split between Ptilocercidae and Tupaiidae occurred *ca* 60 Ma, preceding the emergence of *Dendrogale*, the earliest branching tupaiid, *ca* 35 Ma; divergence of the two extant congeners, *D. melanura* and *D. murina*, is estimated at *ca* 21 Ma [2]. Our results raise the possibility that the common ancestor of *Dendrogale* occupied a distinct visual niche that favoured the fixation of a red-shifted opsin, once this novel mutation arose. One possibility is that *Dendrogale* is adapted to relatively open habitats with greater exposure to unfiltered daylight. Such light is relatively enriched in longer wavelengths compared to foliage-filtered downwelling light [42,43]. In consequence, redder objects should appear brighter in open habitats illuminated with redder light; however, the nature of these putative objects (foods, russet-coloured predators) is uncertain.

Some support for this 'open habitat' hypothesis is evident in the natural history of *D. murina*—it is found in degraded evergreen forest, mixed deciduous forest, bamboo fields without a dicotyledonous canopy, as well as streamside tangles in rocky savannahs [19]. It is also evident in the differences between two sympatric montane species in Mount Kinabalu National Park, Sabah, Borneo [44]. Smooth-tailed treeshrews (*D. melanura*) are found around elevations of well above 900 m, the point at which another species of montane treeshrew, *Tupaia montana*, replaces its lowland congeners *T. gracilis*, *T. longipes*, *T. minor* and *T. tana* [40,44]. The elevational overlap of *D. melanura* and *T. montana* is telling, as the smaller *D. melanura* has a limited range, patchier distribution, and prefers relatively open areas around mossy boulders, whereas *T. montana* prefers greater forest cover and uses a larger range of montane habitats, as reflected by its higher abundance in trap records ([2,40]; K. Wells 2009–2010, personal observation). Importantly, our finding indicates deep antiquity for the red-shifted *OPN1LW* in *Dendrogale*. This interpretation argues against the possibility of recent character displacement; i.e. that the colour vision of *D. melanura* is the result of competition with *T. montana* and niche-divergence in diet, microhabitat or other aspects of visual ecology [15]. However, early opsin divergence between *Dendrogale* and other tupaiids may have facilitated current sympatry via different microhabitat preferences, i.e. habitat patches with less intensive canopy cover versus greener, denser canopy cover, a hypothesis that invites testing through behavioural observation and characterization of downwelling light. Overall, there is a need to better understand fine-scale habitat use by sympatric species at different times of the year and to characterize the light conditions in different microhabitats.

Finally, it is alternatively possible that the $_A180_S$ substitution in *Dendrogale* did not impact the fitness of individuals, but spread through an ancestral population neutrally. If so, this may point to a bottleneck event in the history of this genus, as neutral mutations are unlikely to become fixed in large populations [45]. Future examination of the population genomics of extant *Dendrogale* species and estimations of $N_E$ and past population contractions and expansions may allow us to favour or rule out this hypothesis. Additionally, increased sampling of *OPN1LW* in these species will also allow us to rule out *OPN1LW* polymorphisms, which are common in neotropical primates, but unknown outside of the primate order [15,35,46].

## 5. Conclusion

We present new data on the opsin gene sequences of the northern smooth-tailed treeshrew, *Dendrogale murina*, and the Mindanao treeshrew, *Tupaia everetti*. The gene codings for both short-wavelength sensitive and long-wavelength sensitive opsins are under purifying selection in each species and presumed to be functional. *Dendrogale murina* shared a derived amino acid with its congener, *D. melanura*, at site 180 of the *OPN1LW*, which should cause a shift in the sensitivity towards reddish light. This may indicate the common ancestor of extant *Dendrogale* taxa occupied a relatively more open habitat—richer in light unfiltered by green foliage—and may contribute to present-day niche separation among sympatric diurnal treeshrews. We end by noting that lack of availability of DNA from *Anathana* precluded analysis of the opsins of the remaining genus in the order Scandentia, but in the future this would shed additional light on pressures shaping treeshrew visual ecology.

Ethics. The collection protocol follows the ethical guidelines of the American Society of Mammalogists [47] as well as the guidelines specified in the gratuitous permit issued by the Department of Environment and Natural Resources

(DENR) Office of the Regional Director 13 (GP no. 224, Local Transport permit no. 2015-01). Export of samples was approved by CITES (export permit no. 21691 A-2016 issued by the DENR Biodiversity Management Bureau).

Data accessibility. Nucleotide sequences were deposited in GenBank (accession nos. MH129025, KY825132 and KY825133 (*OPN1SW*), and MH129024, KY825134 and KY825135 (*OPN1LW*)). *OPN1SW* and *OPN1LW* nucleotide and amino acid sequence alignments are available in electronic supplementary material, figure S1 and at the Dryad Digital Repository (https://doi.org/10.5061/dryad.56b1t40) [48].

Authors' contributions. A.D.M. and N.J.D. conceived of the study. G.D. carried out the molecular laboratory work and participated in data analysis; M.C.J. contributed to data analyses; P.S.O. contributed to sampling; G.D., A.D.M. and N.J.D. drafted the manuscript and M.C.J., P.S.O. and K.W. helped in writing the manuscript. All authors gave the final approval for publication.

Competing interests. We have no competing interests.

Funding. This study was funded by National Sciences and Engineering Research Council of Canada, the Canada Research Chairs program and the Alberta Children's Hospital Research Institute to A.D.M. and by the EDC MAGBU to P.S.O. Research in the Philippines was supported by the David and Lucile Packard Foundation (Fellowship in Science and Engineering no. 2007-31754 N.J.D.), the Nelson A. Rockefeller Center, Dartmouth College (Faculty Research grant to N.J.D) and the EDC MAGBU to P.S.O.

Acknowledgements. We dedicate this paper to our coauthor, Professor Perry S. Ong, a champion for Philippine biodiversity and research who passed away on 2 March 2019. We thank Larry Heaney for the constructive comments at the inception of this project, Link Olson for facilitating access to tissue samples, the Department of Environment and Natural Resources (DENR) for permission to collect samples and conduct research and the Energy Development Corporation Mt Apo Geothermal Business Unit (EDC MAGBU) for the permission to work in their geothermal production fields. We thank the research staff of the Biodiversity Research Laboratory Institute of Biology, College of Science, University of the Philippines Diliman (BRL UP Biology) and staff of EDC MAGBU including local guides for conducting the fieldwork.

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
