## [Reviewer comments · Royal Society Open Science]

Review History

RSOS-182037.R0 (Original submission)

Review form: Reviewer 1

Is the manuscript scientifically sound in its present form?

Yes

Are the interpretations and conclusions justified by the results?

Yes

Is the language acceptable?

Yes

Is it clear how to access all supporting data?

Yes

Do you have any ethical concerns with this paper?

No

Have you any concerns about statistical analyses in this paper?

No

Recommendation?

Accept with minor revision (please list in comments)

Comments to the Author(s)

This manuscript makes a significant contribution to our understanding of treeshrew opsin genes and visual ecology & evolution. The subject matter is certainly appropriate for Royal Society Open Science. However, the manuscript must be revised before it can be published, as there are currently fundamental problems with both taxonomic and phylogenetic concepts in the paper. I have listed these issues below.

Title: “Urogale” must be removed from the title. This genus hasn’t been recognized since 2011 (see Roberts et al. 2011), and is not recognized in any modern classification of treeshrews. I encourage the authors to read the following recent taxonomic references:

ASM Mammal Diversity Database: <https://mammaldiversity.org/species-account.php?genus=tupaia&species=everetti>

Burgin CJ, Colella JP, Kahn PL, Upham NS. 2018. How many species of mammals are there? *J Mammal* 99(1):1-14. <https://doi.org/10.1093/jmammal/gyx147>

Hawkins MT. 2018. Family Tupaiidae (Treeshrews). In: Wilson DE, Mittermeier RA, editors. *Handbook of the Mammals of the World Volume 8 Insectivores, Sloths and Colugos*. Barcelona: Lynx Edicions. P 242-269.

Tabaranza, B., Gonzalez, J.C., Rosell-Ambal, R.G.B. & Heaney, L. 2017. *Tupaia everetti* (amended version of 2016 assessment). The IUCN Red List of Threatened Species 2017: e.T22784A114120582. <http://dx.doi.org/10.2305/IUCN.UK.2017-1.RLTS.T22784A114120582.en>.

Furthermore, as the title is currently written, *Tupaia* is listed in parentheses after *Urogale*, which would indicate a subgenus. *Tupaia* Raffles, 1821 cannot be a subgenus of *Urogale* Mearns, 1905 because the former has priority. This is why *Urogale* has been synonymized with *Tupaia*. In short, *Urogale* is no longer recognized, and the authors have indicated a subgenus in the title that violates taxonomic priority. Hence, removing *Urogale* from the title is not optional.

Keywords: “*Urogale*” must be removed from the keywords for the same reasons listed above. Again, this is not optional because *Tupaia* cannot be a subgenus (indicated with parentheses) of *Urogale* (see above).

Summary

p. 2, line 6: Scandentia is not necessarily nested in Euarchonta, as Scandentia is often recovered as the sister to Primatomorpha (Primates + Dermoptera). More accurately, Scandentia “is closely related to” Primates in Euarchonta.

p. 2, line 20: *Tupaia* Raffles, 1821 is not the “junior synonym” of *Urogale* Mearns, 1905 (see above). Quite the opposite. *Tupaia* (senior synonym) has priority, and therefore *Urogale* (junior synonym) has been synonymized with it.

Introduction

p. 3, line 8: no classification is cited for 19 extant tupaiid species. Which outdated classification

are they following? Hawkins (2018, see above) includes 22, and the ASM Mammal Diversity Database (see above) lists 23:

<https://mammaldiversity.org/#dHVwYWlpcZGFJmndsb2JhbF9zZWZyY2g9dHJ1ZSZsb29zZT10cnVI>

p. 3, line 10: see comment about “nested” above. Scandentia is the sister to Primatomorpha in Euarchonta.

p. 3, line 20: the extant *Dendrogale* is not “basal.” It is the sister to the remaining tupaiid taxa. There is a literature on such misinterpretation of “basal,” and I would encourage the authors to consult the following papers:

Crisp MD, Cook LG. 2005. Do early branching lineages signify ancestral traits? *Trends Ecol Evol* 20:122-128. See Figure 1.

Krell FT, Cranston PS. 2004. Which side of the tree is more basal? *Systematic Entomology* 29:279-281.

p. 3, lines 33, 37: change “*Urogale*” to “*Tupaia*” to recognize modern taxonomy (see above).

p. 3, lines 35-37: it’s a mischaracterization to say that only Roberts et al. (2011) have “argued for” the subsumption of *Urogale* into *Tupaia* as this has actually been adopted in every recent (2017-2018) classification of treeshrews (see above). The classification of the Mindanao treeshrew is not “in flux,” and this manuscript on treeshrews should follow modern taxonomy rather than outdated classifications. I don’t see how using “traditional nomenclature” is a “matter of practical convenience” other than failing to update the taxonomy and ignoring recent publications on the topic. The taxonomy should be updated throughout (i.e., use *Tupaia everetti* rather than *Urogale*), and there should be one sentence explaining that this species used to be included in *Urogale*. That will cover the relevant taxonomic history, which is actually more complex (see Lyon 1913).

p. 3, lines 37-39: *T. everetti* is part of *Tupaia*, so it doesn’t occupy a “central position between *Dendrogale* and *Tupaia*.” Do they mean between *Dendrogale* and other *Tupaia* species?

p. 3, line 39: the extant *T. everetti* is not “basal” (see above). Again, there appears to be a fundamental misunderstanding of this term. See Crisp and Cook (2005, Fig. 1).

p. 3, line 47: change “U.” to “T.” to recognize modern taxonomy (see above). This should be done throughout the manuscript.

Materials and Methods

p. 4, line 16: change “*Urogale*” to “*Tupaia*” here and throughout to recognize modern taxonomy (see above)

p. 4, lines 26-27: why rely entirely on captive weights? Sargis (2002) tabled wild-caught weights for “*Urogale*.”

Sargis EJ. 2002. A multivariate analysis of the postcranium of tree shrews (Scandentia, Tupaiidae) and its taxonomic implications. *Mammalia* 66:579-598. Table 4.

p. 4, lines 31-33: UAM is just University of Alaska Museum, so remove “of the North.”

p. 4, line 33: the catalog number is the critical one; the CITES export permit number is unnecessary here and could be removed.

p. 4, line 39: accession numbers are inadequate. Museum catalog numbers must be added.

p. 4, line 41: trapping details (Victor snap-trap baited with earthworms) are unnecessary here and could be removed.

Results

p. 6, line 26: change "U." to "T." to recognize modern taxonomy (see above). Also, "and" should not be italicized. This should be changed to "Relative to *T. everetti* and other *Tupaia* species."

p. 6, lines 30-32: insert "the" between "whether" and "last."

p. 6, line 38: change "U." to "T." to recognize modern taxonomy (see above). Also, "other" should be inserted between "from" and "*Tupaia*."

p. 6, line 40: "may be the ancestral condition" for what? *Tupaiidae*? *Scandentia*? What is the state in *Ptilocercus*?

p. 7, lines 10-12: "We predict that the...to have a similar" must be corrected. Change "to have" to "has."

Discussion

p. 7, line 53: "colour vision of *U. everetti* is practically identical to that of *Tupaia*." Yes, because it IS *Tupaia* (see above). Again, the taxonomy must be updated here. This should be changed to "colour vision of *T. everetti* is practically identical to that of other *Tupaia* species."

p. 8, lines 10-12: the extant *Dendrogale* is not "the basal *tupaiid*" (see above). Again, this term is being used incorrectly. *Dendrogale* is the sister to the remaining *tupaiid* taxa. See Crisp and Cook (2005, Fig. 1) and Krell and Cranston (2004).

p. 8, lines 12-14: "divergence of the two extant congeners, *D. melanura* and *D. murina*, is estimated at 13 Mya (range: 20-5 Mya." This divergence appears to be ~21 Mya, not 13 Mya, in Roberts et al. (2011) Fig. 3.

p. 8, line 14: "last" is unnecessary, as there is only one common ancestor.

p. 8, line 41: "*T. montana*, the more recent immigrant." What is the evidence for recent immigration? A reference should certainly be provided for this.

p. 8, line 43: what does the extra "1" indicate in the references?

p. 8, line 56: remove "along".

p. 9, line 12: "Primate" should not be capitalized. It should be "primate order" in this case. It should only be capitalized when being used as a formal taxonomic name as in the order *Primates*.

Conclusion

p. 9, line 22: "*Urogale*" must be removed for the same reasons listed above. The authors have

proposed an invalid subgenus by including *Tupaia* in parentheses. Again, this change is not optional because *Tupaia* cannot be a subgenus (indicated with parentheses) of *Urogale* (see above).

p. 9, line 28: again, “last” is unnecessary.

References

#16: *Dendrogale murina* must be italicized.

#20: reference is outdated. This IUCN account was updated in 2017 when “*Urogale*” was changed to “*Tupaia*” (see above). The version cited here was accessed two years ago on 2/21/17. This must be updated to the current IUCN version for this 2019 manuscript submission.

Figure 1

p. 13, line 11: again, see comment about “nested” above. Scandentia is the sister to Primatomorpha in Euarchonta.

p. 13, line 13: “*Some authors” is a mischaracterization. Every taxonomist focusing on treeshrews has synonymized *Urogale* with *Tupaia*. See the 3 recent references listed above. The taxonomy being used in this manuscript is out of date and must be updated. Similarly, the literature review is inadequate, and recent taxonomic references must be cited and added to the reference list.

Figure 2

p. 13, line 21: “*Urogale*” must be removed from the caption for the same reasons listed above. Again, this is not optional because *Tupaia* cannot be a subgenus (indicated with parentheses) of *Urogale* (see above).

Figure 3

p. 13, line 34: only reference 15 from 2011 is cited despite starting the sentence with “More recently.” Even more recently are the three 2017-2018 taxonomic references listed above, all of which should be cited here.

Table S1

p. 13, line 43: change “*Urogale*” to “*Tupaia*” to recognize modern taxonomy (see above).

Review form: Reviewer 2

Is the manuscript scientifically sound in its present form?

Yes

Are the interpretations and conclusions justified by the results?

Yes

Is the language acceptable?

Yes

Is it clear how to access all supporting data?

Yes

Do you have any ethical concerns with this paper?

No

Have you any concerns about statistical analyses in this paper?

No

Recommendation?

Accept with minor revision (please list in comments)

Comments to the Author(s)

General comments

Duytschaever et al. sequenced cone opsin genes (SWS1 and M/LWS) from two treeshrew species (*Dendrogale murina* and *Urogale everetti*), inferred the phenotypes of their visual pigments, and then discussed the ecological implications. This is an in-depth study on treeshrew opsins following the authors' previous publication on euarchontans. Importantly, the authors sequenced cone opsin genes from the two key species in order Scandentia, and then reconstructed a better ancestral state of treeshrew color vision. The paper is interesting and will be valuable for researchers who are working on the evolution of treeshrews, euarchontans and also other mammals.

Specific comments

1. It will be better for the authors to add a statement for the site numberings of SWS1 and LWS, which are following bovine RH1 and LWS respectively. Or else, the authors could use the numbering of bovine RH1 for both genes.
2. Page 5, line 57. Specify which branch models used, like free-ratio or one-ratio model.
3. Page 6, line 2. Remove the words "of the tuning". Previous publications already reported that even if the tuning sites are not under positive selection, they are still functionally important for visual pigments. So it's not necessary to test whether the known tuning sites are under positive selection or not. However, it will be interesting to test whether any new sites under positive selection, which will give further candidate(s) for in vitro assay.
4. Page 6, line 49. Change the word SWS to SWS1 to distinguish it from the other short wavelength sensitive opsin SWS2. Also change the word SWS to SWS1 in Figure 3.
5. Page 7, line 18. Show the full name for LR when it first appears.
6. Page 7, line 19. Put the results based on branch models in front of that based on site models, according to Methods.
7. Page 7, line 42. I suggest remove the PROVEAN analyses for both genes, given the results do not help for the phenotypic inferences (λ max) of visual pigments in this story. Moreover, the result contradicts with the phenotypic change of the known critical site (180) for M/LWS and this is hard to explain.

Decision letter (RSOS-182037.R0)

11-Feb-2019

Dear Dr Duytschaever

On behalf of the Editors, I am pleased to inform you that your Manuscript RSOS-182037 entitled "Opsin genes of select treeshrews *Dendrogale murina* and *Urogale (*Tupaia*) everetti* resolve ancestral

character states within *Scandentia*" has been accepted for publication in Royal Society Open Science subject to minor revision in accordance with the referee suggestions. Please find the referees' comments at the end of this email.

The reviewers and handling editors have recommended publication, but note that the reviewers, particularly reviewer 1, require an extensive list of minor revisions to your manuscript. The taxonomy and references must be updated, and some phylogenetic misinterpretations must be corrected. Therefore, I invite you to respond to the comments and revise your manuscript.

- Ethics statement

- Data accessibility

If you wish to submit your supporting data or code to Dryad (<http://datadryad.org/>), or modify your current submission to dryad, please use the following link:
<http://datadryad.org/submit?journalID=RSOS&manu=RSOS-182037>

- Competing interests

- Authors' contributions

- Acknowledgements

- Funding statement

Because the schedule for publication is very tight, it is a condition of publication that you submit the revised version of your manuscript before 20-Feb-2019. Please note that the revision deadline will expire at 00.00am on this date. If you do not think you will be able to meet this date please let me know immediately.

on behalf of Dr Steve Brown (Subject Editor)
openscience@royalsociety.org

Reviewer comments to Author:
Reviewer: 1

Comments to the Author(s)

This manuscript makes a significant contribution to our understanding of treeshrew opsin genes and visual ecology & evolution. The subject matter is certainly appropriate for Royal Society Open Science. However, the manuscript must be revised before it can be published, as there are currently fundamental problems with both taxonomic and phylogenetic concepts in the paper. I have listed these issues below.

Title: "Urogale" must be removed from the title. This genus hasn't been recognized since 2011 (see Roberts et al. 2011), and is not recognized in any modern classification of treeshrews. I encourage the authors to read the following recent taxonomic references:

ASM Mammal Diversity Database: <https://mammaldiversity.org/species-account.php?genus=tupaia&species=everetti>

Burgin CJ, Colella JP, Kahn PL, Upham NS. 2018. How many species of mammals are there? *J Mammal* 99(1):1-14. <https://doi.org/10.1093/jmammal/gyx147>

Hawkins MT. 2018. Family Tupaiidae (Treeshrews). In: Wilson DE, Mittermeier RA, editors. Handbook of the Mammals of the World Volume 8 Insectivores, Sloths and Colugos. Barcelona: Lynx Edicions. P 242-269.

Tabaranza, B., Gonzalez, J.C., Rosell-Ambal, R.G.B. & Heaney, L. 2017. *Tupaia everetti* (amended version of 2016 assessment). The IUCN Red List of Threatened Species 2017: e.T22784A114120582. <http://dx.doi.org/10.2305/IUCN.UK.2017-1.RLTS.T22784A114120582.en>.

Furthermore, as the title is currently written, *Tupaia* is listed in parentheses after *Urogale*, which would indicate a subgenus. *Tupaia* Raffles, 1821 cannot be a subgenus of *Urogale* Mearns, 1905 because the former has priority. This is why *Urogale* has been synonymized with *Tupaia*. In short, *Urogale* is no longer recognized, and the authors have indicated a subgenus in the title that violates taxonomic priority. Hence, removing *Urogale* from the title is not optional.

Keywords: “*Urogale*” must be removed from the keywords for the same reasons listed above. Again, this is not optional because *Tupaia* cannot be a subgenus (indicated with parentheses) of *Urogale* (see above).

Summary

p. 2, line 6: Scandentia is not necessarily nested in Euarchonta, as Scandentia is often recovered as the sister to Primatomorpha (Primates + Dermoptera). More accurately, Scandentia “is closely related to” Primates in Euarchonta.

p. 2, line 20: *Tupaia* Raffles, 1821 is not the “junior synonym” of *Urogale* Mearns, 1905 (see above). Quite the opposite. *Tupaia* (senior synonym) has priority, and therefore *Urogale* (junior synonym) has been synonymized with it.

Introduction

p. 3, line 8: no classification is cited for 19 extant tupaiid species. Which outdated classification are they following? Hawkins (2018, see above) includes 22, and the ASM Mammal Diversity Database (see above) lists 23: <https://mammaldiversity.org/#dHVwYWlpZGFJmidsb2JhbF9zZWZyY2g9dHJ1ZSZsb29zZT10cnVl>

p. 3, line 10: see comment about “nested” above. Scandentia is the sister to Primatomorpha in Euarchonta.

p. 3, line 20: the extant *Dendrogale* is not “basal.” It is the sister to the remaining tupaiid taxa. There is a literature on such misinterpretation of “basal,” and I would encourage the authors to consult the following papers:

Crisp MD, Cook LG. 2005. Do early branching lineages signify ancestral traits? *Trends Ecol Evol* 20:122-128. See Figure 1.

Krell FT, Cranston PS. 2004. Which side of the tree is more basal? *Systematic Entomology* 29:279-281.

p. 3, lines 33, 37: change “*Urogale*” to “*Tupaia*” to recognize modern taxonomy (see above).

p. 3, lines 35-37: it’s a mischaracterization to say that only Roberts et al. (2011) have “argued for”

the subsumption of *Urogale* into *Tupaia* as this has actually been adopted in every recent (2017-2018) classification of treeshrews (see above). The classification of the Mindanao treeshrew is not “in flux,” and this manuscript on treeshrews should follow modern taxonomy rather than outdated classifications. I don’t see how using “traditional nomenclature” is a “matter of practical convenience” other than failing to update the taxonomy and ignoring recent publications on the topic. The taxonomy should be updated throughout (i.e., use *Tupaia everetti* rather than *Urogale*), and there should be one sentence explaining that this species used to be included in *Urogale*. That will cover the relevant taxonomic history, which is actually more complex (see Lyon 1913).

p. 3, lines 37-39: *T. everetti* is part of *Tupaia*, so it doesn’t occupy a “central position between *Dendrogale* and *Tupaia*.” Do they mean between *Dendrogale* and other *Tupaia* species?

p. 3, line 39: the extant *T. everetti* is not “basal” (see above). Again, there appears to be a fundamental misunderstanding of this term. See Crisp and Cook (2005, Fig. 1).

p. 3, line 47: change “U.” to “T.” to recognize modern taxonomy (see above). This should be done throughout the manuscript.

Materials and Methods

p. 4, line 16: change “*Urogale*” to “*Tupaia*” here and throughout to recognize modern taxonomy (see above)

p. 4, lines 26-27: why rely entirely on captive weights? Sargis (2002) tabled wild-caught weights for “*Urogale*.”

Sargis EJ. 2002. A multivariate analysis of the postcranium of tree shrews (Scandentia, Tupaiidae) and its taxonomic implications. *Mammalia* 66:579-598. Table 4.

p. 4, lines 31-33: UAM is just University of Alaska Museum, so remove “of the North.”

p. 4, line 33: the catalog number is the critical one; the CITES export permit number is unnecessary here and could be removed.

p. 4, line 39: accession numbers are inadequate. Museum catalog numbers must be added.

p. 4, line 41: trapping details (Victor snap-trap baited with earthworms) are unnecessary here and could be removed.

Results

p. 6, line 26: change “U.” to “T.” to recognize modern taxonomy (see above). Also, “and” should not be italicized. This should be changed to “Relative to *T. everetti* and other *Tupaia* species.”

p. 6, lines 30-32: insert “the” between “whether” and “last.”

p. 6, line 38: change “U.” to “T.” to recognize modern taxonomy (see above). Also, “other” should be inserted between “from” and “*Tupaia*.”

p. 6, line 40: “may be the ancestral condition” for what? Tupaiidae? Scandentia? What is the state in *Ptilocercus*?

p. 7, lines 10-12: “We predict that the...to have a similar” must be corrected. Change “to have” to “has.”

Discussion

p. 7, line 53: “colour vision of *U. everetti* is practically identical to that of *Tupaia*.” Yes, because it IS *Tupaia* (see above). Again, the taxonomy must be updated here. This should be changed to “colour vision of *T. everetti* is practically identical to that of other *Tupaia* species.”

p. 8, lines 10-12: the extant *Dendrogale* is not “the basal tupaiid” (see above). Again, this term is being used incorrectly. *Dendrogale* is the sister to the remaining tupaiid taxa. See Crisp and Cook (2005, Fig. 1) and Krell and Cranston (2004).

p. 8, lines 12-14: “divergence of the two extant congeners, *D. melanura* and *D. murina*, is estimated at 13 Mya (range: 20-5 Mya.” This divergence appears to be ~21 Mya, not 13 Mya, in Roberts et al. (2011) Fig. 3.

p. 8, line 14: “last” is unnecessary, as there is only one common ancestor.

p. 8, line 41: “*T. montana*, the more recent immigrant.” What is the evidence for recent immigration? A reference should certainly be provided for this.

p. 8, line 43: what does the extra “1” indicate in the references?

p. 8, line 56: remove “along”.

p. 9, line 12: “Primate” should not be capitalized. It should be “primate order” in this case. It should only be capitalized when being used as a formal taxonomic name as in the order Primates.

Conclusion

p. 9, line 22: “*Urogale*” must be removed for the same reasons listed above. The authors have proposed an invalid subgenus by including *Tupaia* in parentheses. Again, this change is not optional because *Tupaia* cannot be a subgenus (indicated with parentheses) of *Urogale* (see above).

p. 9, line 28: again, “last” is unnecessary.

References

#16: *Dendrogale murina* must be italicized.

#20: reference is outdated. This IUCN account was updated in 2017 when “*Urogale*” was changed to “*Tupaia*” (see above). The version cited here was accessed two years ago on 2/21/17. This must be updated to the current IUCN version for this 2019 manuscript submission.

Figure 1

p. 13, line 11: again, see comment about “nested” above. Scandentia is the sister to Primatomorpha in Euarchonta.

p. 13, line 13: “*Some authors” is a mischaracterization. Every taxonomist focusing on treeshrews

has synonymized *Urogale* with *Tupaia*. See the 3 recent references listed above. The taxonomy being used in this manuscript is out of date and must be updated. Similarly, the literature review is inadequate, and recent taxonomic references must be cited and added to the reference list.

Figure 2

p. 13, line 21: “*Urogale*” must be removed from the caption for the same reasons listed above. Again, this is not optional because *Tupaia* cannot be a subgenus (indicated with parentheses) of *Urogale* (see above).

Figure 3

p. 13, line 34: only reference 15 from 2011 is cited despite starting the sentence with “More recently.” Even more recently are the three 2017-2018 taxonomic references listed above, all of which should be cited here.

Table S1

p. 13, line 43: change “*Urogale*” to “*Tupaia*” to recognize modern taxonomy (see above).

Reviewer: 2

Comments to the Author(s)

General comments

Duytschaever et al. sequenced cone opsin genes (SWS1 and M/LWS) from two treeshrew species (*Dendrogale murina* and *Urogale everetti*), inferred the phenotypes of their visual pigments, and then discussed the ecological implications. This is an in-depth study on treeshrew opsins following the authors' previous publication on euarchontans. Importantly, the authors sequenced cone opsin genes from the two key species in order Scandentia, and then reconstructed a better ancestral state of treeshrew color vision. The paper is interesting and will be valuable for researchers who are working on the evolution of treeshrews, euarchontans and also other mammals.

Specific comments

1. It will be better for the authors to add a statement for the site numberings of SWS1 and LWS, which are following bovine RH1 and LWS respectively. Or else, the authors could use the numbering of bovine RH1 for both genes.
2. Page 5, line 57. Specify which branch models used, like free-ratio or one-ratio model.
3. Page 6, line 2. Remove the words "of the tuning". Previous publications already reported that even if the tuning sites are not under positive selection, they are still functionally important for visual pigments. So it's not necessary to test whether the known tuning sites are under positive selection or not. However, it will be interesting to test whether any new sites under positive selection, which will give further candidate(s) for in vitro assay.
4. Page 6, line 49. Change the word SWS to SWS1 to distinguish it from the other short wavelength sensitive opsin SWS2. Also change the word SWS to SWS1 in Figure 3.
5. Page 7, line 18. Show the full name for LR when it first appears.
6. Page 7, line 19. Put the results based on branch models in front of that based on site models, according to Methods.
7. Page 7, line 42. I suggest remove the PROVEAN analyses for both genes, given the results do not help for the phenotypic inferences (λ max) of visual pigments in this story. Moreover, the result contradicts with the phenotypic change of the known critical site (180) for M/LWS and this is hard to explain.

Author's Response to Decision Letter for (RSOS-182037.R0)

See Appendix A.

Decision letter (RSOS-182037.R1)

21-Mar-2019

Dear Dr Duytschaever,

I am pleased to inform you that your manuscript entitled "Opsin genes of select treeshrews resolve ancestral character states within Scandentia" is now accepted for publication in Royal Society Open Science.

on behalf of Dr Steve Brown (Associate Editor) and Professor Steve Brown (Subject Editor)
openscience@royalsociety.org

Appendix A

Manuscript ID RSOS-182037

ID RSOS-182037 Duytschaever et al. Response to Referees

Editors: Andrew Dunn, Dr. Steve Brown

On behalf of the Editors, I am pleased to inform you that your Manuscript RSOS-182037 entitled "Opsin genes of select treeshrews *Dendrogale murina* and *Urogale* (*Tupaia*) *everetti* resolve ancestral character states within Scandentia" has been accepted for publication in Royal Society Open Science subject to minor revision in accordance with the referee suggestions. Please find the referees' comments at the end of this email.

The reviewers and handling editors have recommended publication, but note that the reviewers, particularly reviewer 1, require an extensive list of minor revisions to your manuscript. The taxonomy and references must be updated, and some phylogenetic misinterpretations must be corrected. Therefore, I invite you to respond to the comments and revise your manuscript.

>>> We thank the reviewers and handling editors for the supportive comments and we are grateful you are considering our work for publication. We have carefully revised our manuscript and updated the taxonomy according to the modern classification of treeshrews. We provide a response to each comment below in blue preceded by ">>>".

Reviewer comments to Author:

Reviewer: 1

Comments to the Author(s)

This manuscript makes a significant contribution to our understanding of treeshrew opsin genes and visual ecology & evolution. The subject matter is certainly appropriate for Royal Society Open Science. However, the manuscript must be revised before it can be published, as there are currently fundamental problems with both taxonomic and phylogenetic concepts in the paper. I have listed these issues below.

>>> We thank Reviewer 1 for carefully reviewing our manuscript and helpful comments. We acknowledge the taxonomic concerns and have revised our manuscript according to the modern treeshrew taxonomy.

Title: "*Urogale*" must be removed from the title. This genus hasn't been recognized since 2011 (see Roberts et al. 2011), and is not recognized in any modern classification of treeshrews. I encourage the authors to read the following recent taxonomic references:

>>> We have altered the title, including removing *Urogale*.

ASM Mammal Diversity Database: <https://mammaldiversity.org/species-account.php?genus=tupaia&species=everetti>

Burgin CJ, Colella JP, Kahn PL, Upham NS. 2018. How many species of mammals are there? J Mammal 99(1):1-14. <https://doi.org/10.1093/jmammal/gyx147>

Hawkins MT. 2018. Family Tupaiidae (Treeshrews). In: Wilson DE, Mittermeier RA, editors. Handbook of the Mammals of the World Volume 8 Insectivores, Sloths and Colugos. Barcelona: Lynx Edicions. P 242-269.

Tabaranza, B., Gonzalez, J.C., Rosell-Ambal, R.G.B. & Heaney, L. 2017. *Tupaia everetti* (amended version of 2016 assessment). The IUCN Red List of Threatened Species 2017: e.T22784A114120582. <http://dx.doi.org/10.2305/IUCN.UK.2017-1.RLTS.T22784A114120582.en>.

Furthermore, as the title is currently written, *Tupaia* is listed in parentheses after *Urogale*, which would indicate a subgenus. *Tupaia* Raffles, 1821 cannot be a subgenus of *Urogale* Mearns, 1905 because the former has priority. This is why *Urogale* has been synonymized with *Tupaia*. In short, *Urogale* is no longer recognized, and the authors have indicated a subgenus in the title that violates taxonomic priority. Hence, removing *Urogale* from the title is not optional.

>>> We thank the reviewer for thoroughly explaining the taxonomic issue. We recognize that *Urogale* has been synonymized with *Tupaia* and that the latter nomen is the senior synonym. To clarify the reversed taxonomy, we have revised the second paragraph in the introduction. We have implemented the suggested nomenclature change from *Urogale everetti* to *Tupaia everetti*. Accordingly we have made changes to the title **Opsin genes of select treeshrews resolve ancestral character states within Scandentia** and throughout the manuscript. In addition, we have read the suggested references and updated our citations.

Keywords: “*Urogale*” must be removed from the keywords for the same reasons listed above. Again, this is not optional because *Tupaia* cannot be a subgenus (indicated with parentheses) of *Urogale* (see above).

>>> Thank you. We have now added *Tupaia everetti* to the keywords. We have left the genus *Urogale* in the list of keywords for index-search purposes. (Persons unfamiliar with the name change reading past work might use this as a search term.)

Summary

p. 2, line 6: Scandentia is not necessarily nested in Euarchonta, as Scandentia is often recovered as the sister to Primatomorpha (Primates + Dermoptera). More accurately, Scandentia “is closely related to” Primates in Euarchonta.

>>> Thank you for your comment. We agree that it is more appropriate to refer to Scandentia as closely related to Primates and Dermoptera in the superordinal group Euarchonta. We have now clarified this in the text. (p. 2, lines 3-4)

p. 2, line 20: *Tupaia* Raffles, 1821 is not the “junior synonym” of *Urogale* Mearns, 1905 (see above). Quite the opposite. *Tupaia* (senior synonym) has priority, and therefore *Urogale* (junior synonym) has been synonymized with it.

>>> We agree with this comment and have altered the text to “*Tupaia everetti*, the senior synonym of *Urogale everetti*.” (p. 2, lines 10-11)

Introduction

p. 3, line 8: no classification is cited for 19 extant tupaiid species. Which outdated classification are they following? Hawkins (2018, see above) includes 22, and the ASM Mammal Diversity Database (see above) lists 23:

<https://mammaldiversity.org/#dHVwYWlpZGFJmJm5b2JhbF9zZWZyY2g9dHJ1ZSZsb29zZT10cnVI>

>>> We thank the reviewer for bringing this to our attention. We have updated the number of species to 22 and added the reference to Hawkins 2018. (p. 3, lines 18-20).

p. 3, line 10: see comment about “nested” above. Scandentia is the sister to Primatomorpha in Euarchonta.

>>> Thank you. We agree that Scandentia is the sister to Primatomorpha and have now clarified this in the text. (p. 3, lines 5-7)

p. 3, line 20: the extant *Dendrogale* is not “basal.” It is the sister to the remaining tupaiid taxa. There is a literature on such misinterpretation of “basal,” and I would encourage the authors to consult the following papers:

Crisp MD, Cook LG. 2005. Do early branching lineages signify ancestral traits? *Trends Ecol Evol* 20:122-128. See Figure 1.

Krell FT, Cranston PS. 2004. Which side of the tree is more basal? *Systematic Entomology* 29:279-281.

>>> Thank you for highlighting this important misinterpretation. We agree with the reviewer and have reworded the sentence to “earliest branching tupaiid”. (p. 3, line 10)

p. 3, lines 33, 37: change “*Urogale*” to “*Tupaia*” to recognize modern taxonomy (see above).

>>> Thank you. We have made the suggested change.

p. 3, lines 35-37: it’s a mischaracterization to say that only Roberts et al. (2011) have “argued for” the subsumption of *Urogale* into *Tupaia* as this has actually been adopted in every recent (2017-2018) classification of treeshrews (see above). The classification of the Mindanao treeshrew is not “in flux,” and this manuscript on treeshrews should follow modern taxonomy rather than outdated classifications. I don’t see how using “traditional nomenclature” is a “matter of practical convenience” other than failing to update the taxonomy and ignoring recent publications on the topic. The taxonomy should be updated throughout (i.e., use *Tupaia everetti* rather than *Urogale*), and there should be one sentence explaining that this species used to be included in *Urogale*. That will cover the relevant taxonomic history, which is actually more complex (see Lyon 1913).

>>> We thank the reviewer for this comment. We acknowledge that the subsumption of *Urogale* into *Tupaia* is now recognized in the modern classification of treeshrews and have accordingly revised this paragraph. We have updated our manuscript according to the modern taxonomy.

p. 3, lines 37-39: *T. everetti* is part of *Tupaia*, so it doesn't occupy a "central position between *Dendrogale* and *Tupaia*." Do they mean between *Dendrogale* and other *Tupaia* species?
>>> Thank you for bringing this to our attention. We recognize that "central position" is a misleading term because *T. everetti* much more closely related to the other tupaids than to *Dendrogale*. We have revised the text as follows to clarify this point:

(p. 3, lines 16-21) Here we fill two voids by focusing on the only extant congener of *Dendrogale melanura*, the northern smooth-tailed treeshrew (*D. murina*), and the Mindanao treeshrew, a species described as *Tupaia everetti* in 1892 and elevated to a monotypic genus (*Urogale*) in 1905 on the basis of distinguishing morphological traits [13]. This latter nomen prevailed for a century until mounting molecular evidence favored the subsumption of *Urogale* into *Tupaia* [12-15]. *Tupaia everetti* is therefore the senior synonym of *U. everetti*, and re-recognition of *T. everetti* is spreading in the literature [1,14].

p. 3, line 39: the extant *T. everetti* is not "basal" (see above). Again, there appears to be a fundamental misunderstanding of this term. See Crisp and Cook (2005, Fig. 1).
>>> We recognize this misinterpretation and have revised the text.

p. 3, line 47: change "U." to "T." to recognize modern taxonomy (see above). This should be done throughout the manuscript.
>>> Thank you. *Urogale* has now been changed to *Tupaia* throughout the manuscript.

Materials and Methods

p. 4, line 16: change "Urogale" to "Tupaia" here and throughout to recognize modern taxonomy (see above)
>>> Thank you. *Urogale* has now been changed to *Tupaia* throughout the manuscript.

p. 4, lines 26-27: why rely entirely on captive weights? Sargis (2002) tabled wild-caught weights for "Urogale:"
Sargis EJ. 2002. A multivariate analysis of the postcranium of tree shrews (Scandentia, Tupaiidae) and its taxonomic implications. *Mammalia* 66:579-598. Table 4.
>>> Thank you for pointing this out. We have reviewed the suggested reference and revised our text accordingly. (p. 4, lines 19-22)

p. 4, lines 31-33: UAM is just University of Alaska Museum, so remove "of the North."
>>> We apologize for this error. We have now removed "of the North".

p. 4, line 33: the catalog number is the critical one; the CITES export permit number is unnecessary here and could be removed.
>>> Thank you. We have removed the CITES export permit information.

p. 4, line 39: accession numbers are inadequate. Museum catalog numbers must be added.

>>> We use accession numbers following the reporting standard used by the University of the Philippines.

p. 4, line 41: trapping details (Victor snap-trap baited with earthworms) are unnecessary here and could be removed.

>>> We have removed the trapping details.

Results

p. 6, line 26: change “U.” to “T.” to recognize modern taxonomy (see above). Also, “and” should not be italicized. This should be changed to “Relative to *T. everetti* and other *Tupaia* species.”

>>> Thank you. We have made the suggested changes.

p. 6, lines 30-32: insert “the” between “whether” and “last.”

>>> Thank you for pointing out this typo. We have added “the” between “whether” and “last”.

p. 6, line 38: change “U.” to “T.” to recognize modern taxonomy (see above). Also, “other” should be inserted between “from” and “*Tupaia*.”

>>> Thank you. We have made the suggested changes.

p. 6, line 40: “may be the ancestral condition” for what? *Tupaia*idae? Scandentia? What is the state in *Ptilocercus*?

>>> We thank the reviewer for allowing us to clarify. We suggest that 28L may be the ancestral condition for *Tupaia*idae and possibly for Scandentia because it is shared with *D. melanura*. We have revised this sentence in the manuscript. P. 7, lines 5-7 “This may be the ancestral condition for *Tupaia*idae and possibly for Scandentia, as it is shared with *D. melanura*. Data at these sites from *Ptilocercus lowii* would help to further resolve this but are presently unavailable.”

p. 7, lines 10-12: “We predict that the...to have a similar” must be corrected. Change “to have” to “has.”

>>> Thank you. We have changed “to have” to “has.”

Discussion

p. 7, line 53: “colour vision of *U. everetti* is practically identical to that of *Tupaia*.” Yes, because it IS *Tupaia* (see above). Again, the taxonomy must be updated here. This should be changed to “colour vision of *T. everetti* is practically identical to that of other *Tupaia* species.”

>>> Thank you for this comment. We recognize that *U. everetti* is *Tupaia everetti* and have changed the text following the reviewer’s suggestion.

p. 8, lines 10-12: the extant *Dendrogale* is not “the basal *tupaiaid*” (see above). Again, this term is being used incorrectly. *Dendrogale* is the sister to the remaining *tupaiaid* taxa. See Crisp and Cook (2005, Fig. 1) and Krell and Cranston (2004).

>>> We agree that *Dendrogale* is not the basal tupaiid and have reworded the sentence to “earliest branching tupaiid”.

p. 8, lines 12-14: “divergence of the two extant congeners, *D. melanura* and *D. murina*, is estimated at 13 Mya (range: 20-5 Mya.” This divergence appears to be ~21 Mya, not 13 Mya, in Roberts et al. (2011) Fig. 3.

>>> We thank the reviewer for bringing this to our attention. We have updated this to 21 Mya and now cite Roberts et al. (2011).

p. 8, line 14: “last” is unnecessary, as there is only one common ancestor.

>>> Thank you. We have removed “last”.

p. 8, line 41: “*T. montana*, the more recent immigrant.” What is the evidence for recent immigration? A reference should certainly be provided for this.

>>> The immigration times of these two genera to a montane habit is not well documented. We have removed the reference to *T. montana* being a more recent immigrant.

p. 8, line 43: what does the extra “1” indicate in the references?

>>> Thank you for pointing out this typo. We have removed the extra “1”.

p. 8, line 56: remove “along”.

>>> Thank you. We have removed “along”.

p. 9, line 12: “Primate” should not be capitalized. It should be “primate order” in this case. It should only be capitalized when being used as a formal taxonomic name as in the order Primates.

>>> We apologize, we have corrected this error.

Conclusion

p. 9, line 22: “*Urogale*” must be removed for the same reasons listed above. The authors have proposed an invalid subgenus by including *Tupaia* in parentheses. Again, this change is not optional because *Tupaia* cannot be a subgenus (indicated with parentheses) of *Urogale* (see above).

>>> We recognize that *U. everetti* is *Tupaia everetti* and have removed *Urogale* following the reviewer’s suggestion.

p. 9, line 28: again, “last” is unnecessary.

>>> Thank you. We have removed “last”.

References

#16: *Dendrogale murina* must be italicized.

>>> Thank you. We have made this change.

#20: reference is outdated. This IUCN account was updated in 2017 when “Urogale” was changed to “Tupaia” (see above). The version cited here was accessed two years ago on 2/21/17. This must be updated to the current IUCN version for this 2019 manuscript submission.

>>> Thank you. We have updated this reference.

Figure 1

p. 13, line 11: again, see comment about “nested” above. Scandentia is the sister to Primatomorpha in Euarchonta.

>>> We agree that Scandentia is the sister to Primatomorpha in Euarchonta and have revised the figure legend accordingly.

p. 13, line 13: “*Some authors” is a mischaracterization. Every taxonomist focusing on treeshrews has synonymized Urogale with Tupaia. See the 3 recent references listed above. The taxonomy being used in this manuscript is out of date and must be updated. Similarly, the literature review is inadequate, and recent taxonomic references must be cited and added to the reference list.

>>> We thank the reviewer for this comment. We have revised the caption and updated our references following the reviewer’s suggestions. We now cite Hawkins (2018) and Tabaranza et al. (2017).

Figure 2

p. 13, line 21: “Urogale” must be removed from the caption for the same reasons listed above. Again, this is not optional because Tupaia cannot be a subgenus (indicated with parentheses) of Urogale (see above).

>>> Thank you. We have removed *Urogale* from the caption.

Figure 3

p. 13, line 34: only reference 15 from 2011 is cited despite starting the sentence with “More recently.” Even more recently are the three 2017-2018 taxonomic references listed above, all of which should be cited here.

>>> We have removed the taxonomy statement from the caption.

Table S1

p. 13, line 43: change “Urogale” to “Tupaia” to recognize modern taxonomy (see above).

>>> Thank you. We have replaced *Urogale* with *Tupaia* in all our tables and figures following the modern taxonomy.

Reviewer: 2

Comments to the Author(s)

General comments

Duytschaever et al. sequenced cone opsin genes (*SWS1* and *M/LWS*) from two treeshrew species (*Dendrogale murina* and *Urogale everetti*), inferred the phenotypes of their visual pigments, and then discussed the ecological implications. This is an in-depth study on treeshrew opsins following the authors' previous publication on euarchontans. Importantly, the authors sequenced cone opsin genes from the two key species in order Scandentia, and then reconstructed a better ancestral state of treeshrew color vision. The paper is interesting and will be valuable for researchers who are working on the evolution of treeshrews, euarchontans and also other mammals.

>>> We thank Reviewer 2 for the positive comments and helpful suggestions.

Specific comments

1. It will be better for the authors to add a statement for the site numberings of *SWS1* and *LWS*, which are following bovine RH1 and *LWS* respectively. Or else, the authors could use the numbering of bovine RH1 for both genes.

>>> We thank the reviewer for this suggestion. A statement indicating the site numbers is included in the caption of figure S1. Site numbers in our alignments correspond to the position of the amino acid in the human *SWS1* and *M/LWS* pigments. We have now included a statement in our manuscript under paragraph "Data analysis" on page 6, lines 6-7.

2. Page 5, line 57. Specify which branch models used, like free-ratio or one-ratio model.

>>> Thank you. For the branch models, we used a free-ratio model. We now specify this in our methods section.

3. Page 6, line 2. Remove the words "of the tuning". Previous publications already reported that even if the tuning sites are not under positive selection, they are still functionally important for visual pigments. So it's not necessary to test whether the known tuning sites are under positive selection or not. However, it will be interesting to test whether any new sites under positive selection, which will give further candidate(s) for in vitro assay.

>>> Thank you for your comment. We agree that even if the tuning sites are not under positive selection, they are still functionally important for visual pigments, but we still wish to be clear that we examined them for positive selection. We did not detect evidence for selection acting on any sites. We have edited the text to clarify our methods and results.

P. 6, lines 15-16 "We also used codeml site models to test whether any sites, including the tuning sites, are under positive selection."

P. 8, lines 1-2 "We did not find evidence of positively selected sites in our codeml site models for either *OPN1SW* ($p = 1$, $LR = 0$) or *OPN1LW* ($p = 0.228$, $LR = 2.957$)."

4. Page 6, line 49. Change the word SWS to SWS1 to distinguish it from the other short wavelength sensitive opsin SWS2. Also change the word SWS to SWS1 in Figure 3.

>>> Thank you for pointing this out. We have made the suggested changes in our manuscript and in Figure 3.

5. Page 7, line 18. Show the full name for LR when it first appears.

>>> LR refers to likelihood ratio. We have now defined this abbreviation in our manuscript. (p. 7, line 28).

6. Page 7, line 19. Put the results based on branch models in front of that based on site models, according to Methods.

>>> Thank you. We have made the suggested change.

7. Page 7, line 42. I suggest remove the PROVEAN analyses for both genes, given the results do not help for the phenotypic inferences (λ_{\max}) of visual pigments in this story. Moreover, the result contradicts with the phenotypic change of the known critical site (180) for M/LWS and this is hard to explain.

>>> Thank you for raising this point. While we acknowledge that our PROVEAN analyses did not predict any deleterious effects of the nonsynonymous amino acid substitutions, we still believe these results are relevant to mention in this paper. When the properties of the amino acid do not change drastically (e.g. due to the amino acid's hydrophobicity), the functional effect of nonsynonymous mutations that change the amino acid will be predicted neutral. The PROVEAN results do not contradict the phenotypic change at site 180, but indicate that the nonsynonymous substitutions are not deleterious to the protein function.